# Weak signal enhancement by nonlinear resonance control in a forced nano-electromechanical resonator

Avishek Chowdhury [1], Marcel G. Clerc[2], Sylvain Barbay [1], Isabelle Robert-Philip[3] & Remy Braive [1,4 ✉]

Driven non-linear resonators can display sharp resonances or even multistable behaviours amenable to induce strong enhancements of weak signals. Such enhancements can make use of the phenomenon of vibrational resonance, whereby a weak low-frequency signal applied to a bistable resonator can be amplified by driving the non-linear oscillator with another appropriately-adjusted non-resonant high-frequency field. Here we demonstrate experimentally and theoretically a significant resonant enhancement of a weak signal by use of a vibrational force, yet in a monostable system consisting of a driven nano-electromechanical nonlinear resonator. The oscillator is subjected to a strong quasi-resonant drive and to two additional tones: a weak signal at lower frequency and a non-resonant driving at an intermediate frequency. We analyse this phenomenon in terms of coherent nonlinear resonance manipulation. Our results illustrate a general mechanism which might have applications in the fields of microwave signal amplification or sensing for instance.

[1] Centre de Nanosciences et de Nanotechnologies, CNRS, Université Paris-Saclay, 10 Boulevard Thomas Gobert, Palaiseau, France. [2] Departamento de Física and Millennium Institute for Research in Optics, Facultad de Ciencias Físicas y Matemáticas, Universidad de Chile, Casilla 487-3 Santiago, Chile. [3] Laboratoire Charles Coulomb, Université de Montpellier, CNRS, 34000 Montpellier, France. [4] Université de Paris, 5 Rue Thomas Mann, 75013 Paris, France. ✉email: remy.braive@c2n.upsaclay.fr

In bistable systems, weak periodic signals can be amplified by use of external driving. Such external driving can be some noise of appropriate strength in the case of stochastic resonance[1], or a high-frequency harmonic signal of appropriate amplitude in the case of vibrational resonance[2]. Both physical phenomena share qualitative features including a resonant-like behaviour, though the underlying mechanisms differ. Time matching criterion dependent on the applied noise amplitude required for stochastic resonance is replaced, in the case of vibrational resonance, by an amplitude criterion equivalently to a parametric amplification near the critical point. Both phenomena have been reported in many different areas including electronics[3,4], optics[5–8] or neurobiology[9,10]. In nanomechanics, the bistable system is usually a simple nonlinear resonator and bistability arises thanks to a quasi-resonant forcing.

This phenomenon can be fully understood thanks to the ubiquitous Duffing model which, beyond nanomechanics, can be used for superconducting Josephson amplifier[11], ionisation waves in plasma[12,13] to describe complex spatiotemporal behaviours such as chimera states[14]. In the frame of the well-known Duffing model, the oscillator features two equilibrium states of different amplitudes and phases for the same values of parameters. In this regime, substantial resonant enhancement of a weak and slowly modulated signal through stochastic resonance can be achieved either by use of amplitude[15–17] or phase[18] noise. When the external driving is no more stochastic but rather a harmonic signal of high frequency, a little bit of care has to be taken. The system is then subjected to forces occurring on three different timescales: the one of the signal, the one of the external drive and the one of the forcing. In the standard picture of vibrational resonance, the signal must have a much smaller frequency than the one of the external drive. We here show an enhancement by a factor up to 20 of a weak modulated signal thanks to vibrational resonance. Moreover, the occurence of vibrational resonance in a forced system requires the external driving frequency not only to be higher than the signal frequency but also to be lower than the forcing frequency. Most importantly, we show in that case that the high-frequency driving amplitude renormalises the forced nonlinear resonator response through the manipulation of the nonlinear resonance. We argue that this effect could be used besides the one we are presenting here, as a general mechanism for nonlinear resonance manipulation. Beyond these fundamental aspects, potential applications encompass various fields such as telecommunications, where the data are encoded on a low-frequency modulation applied to a high-frequency carrier, or, more prospectively, microwave signals processing and sensing such as accelerometers featuring enhanced sensitivities via optomechanically driven signal amplification[19] or for torque magnetometry[20]. Bichromatic signals are in addition pervasive in many other fields including brain-inspired architecture mimicking neural networks[21] where bursting neurons may exhibit two widely different time scales. Finally, enhancement of weak signal by vibrational resonance might be valuable for binary logic gate based on phase transition for reprogrammable logic operation[22,23] as well as for memory operation[24], physical simulators[25,26] and more prospective application as quantum computing[27].

## Results

**Nonlinear driving of the nano-electromechanical resonator.** In our experiments, the resonator consists of a non-linear nano-electromechanical oscillator formed by a thin micron-scale InP suspended membrane (see Fig. 1a). The membrane's out-of-plane motion is induced by applying an AC voltage $V(t)$ on integrated metallic interdigitated electrodes placed underneath the membrane at a sub-micron distance (see Fig. 1a; see Chowdhury et al.[28] and Methods for more details). It is placed in a low-pressure chamber ($10^{-4}$ mbar) in order to reduce mechanical damping. The out-of-plane motion of the membrane is probed optically (see Fig. 1b) thanks to a Michelson interferometer whose one end mirror is formed by the oscillating membrane (see Fig. 1a). The membrane mechanical fundamental mode of oscillation lies at 2.82 MHz in the linear regime with a mechanical quality factor of $Q_M \sim 10^3$.

The AC forcing voltage lies in the MHz regime and writes:

$$V(t) = V_0\cos[2\pi\nu_f t] \qquad (1)$$

Here $V_0$ is the amplitude of the applied voltage while $\nu_f$ denotes the frequency of the quasi-resonant forcing. When sweeping up and down the frequency $\nu_f$ in the vicinity of the fundamental mode frequency, asymmetry in the mechanical response spectrum appears for a sufficiently high driving amplitude $V_0 > 5.5V$ (see Fig. 1b). Hysteresis behaviour becomes prominent and two stable points, represented by the low and the high amplitude values of the mechanical motion, co-exist. The evolution of the bistable region width as a function of $V_0$ is shown in Fig. 1b Right. The closing of this bistable region for increasing $V_0$ cannot be

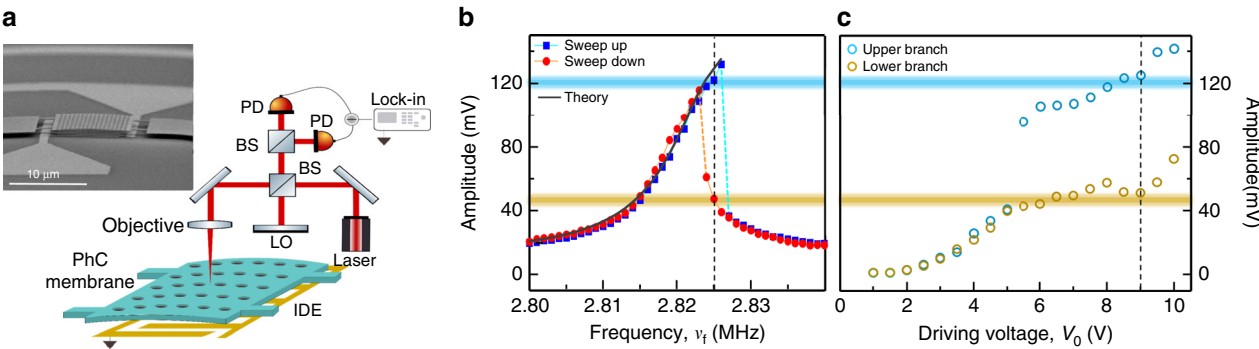

**Fig. 1 Nonlinear characterization of the nano-electromechanical resonator. a** Schematic of the experimental set-up used for the actuation and detection of the mechanical motion of the membrane (PhC photonic crystal, IDE interdigitated electrodes, LO local oscillator, BS beamsplitter, PD photodiode). Inset: scanning electron microscopic image of the resonator consisting of a suspended InP membrane with a thickness of 260 nm and a surface of $20 \times 10$ μm$^2$ and integrating gold interdigitated electrodes underneath at a 400-nm distance. **b** Amplitude of the forced mechanical fundamental mode as a function of the forcing frequency $\nu_f$ in a sweep up (blue dots) and sweep down (red dots) experiment for $V_0 = 9V$. A theoretical fitting of the forced amplitude response curve (sweep up) is shown (black line). The vertical dashed line lies at a frequency of 2.825 MHz. **c** Amplitude of the forced mechanical fundamental mode as a function of the amplitude of the applied voltage $V_0$ for a forcing frequency fixed at $\nu_f = 2.825$ MHz. The vertical dashed line lies at a voltage amplitude of $9V$.

observed due to limited voltage handled at the electrodes terminals. In the following, $V_0$ will be set to $9V$ in order to be deeply in the bistable regime, and the forcing frequency is set at $\nu_f = 2.825\,\text{MHz}$, close to the middle of the hysteresis region, in order to get symmetrical potentials[18].

**Externally induced dynamics in the time domain.** Jumps between the two stable states of oscillation can be induced by slowly modulating the forcing amplitude. This scenario can be implemented by applying to the electrodes a voltage in the form of:

$$V(t) = V_0(1 + \gamma \cos(2\pi\nu_m t)) \cdot \cos[2\pi\nu_f t] \quad (2)$$

where $\gamma$ and $\nu_m$ denote respectively the modulation index and the frequency of the amplitude modulated signal. Yet, a sufficiently high modulation amplitude is needed to drive the system in order to overcome the barrier height and to induce inter-well motion following the applied modulation. In the case of a weak amplitude modulated signal (as in our experiments with $\gamma$ set at 0.1), the system is solely subjected to intra-well modulated motion as can be seen in Fig. 2 top for the resonator being initially prepared in its upper state. Amplification of the weak modulated signal following jumps of the system between the two states can however still be induced in that case by adding an external driving with a frequency that is much higher than the frequency of the weak modulation, but still lower than the forcing frequency.

In this scenario reproduced in our experiment, the external drive takes the form of an additive amplitude modulation voltage of amplitude $V_{HF} \equiv \delta * V_0$ and frequency $\nu_{HF}$. The criteria for enabling the onset of vibrational resonance as predicted by theory (see section theoretical analysis), requires strong frequency

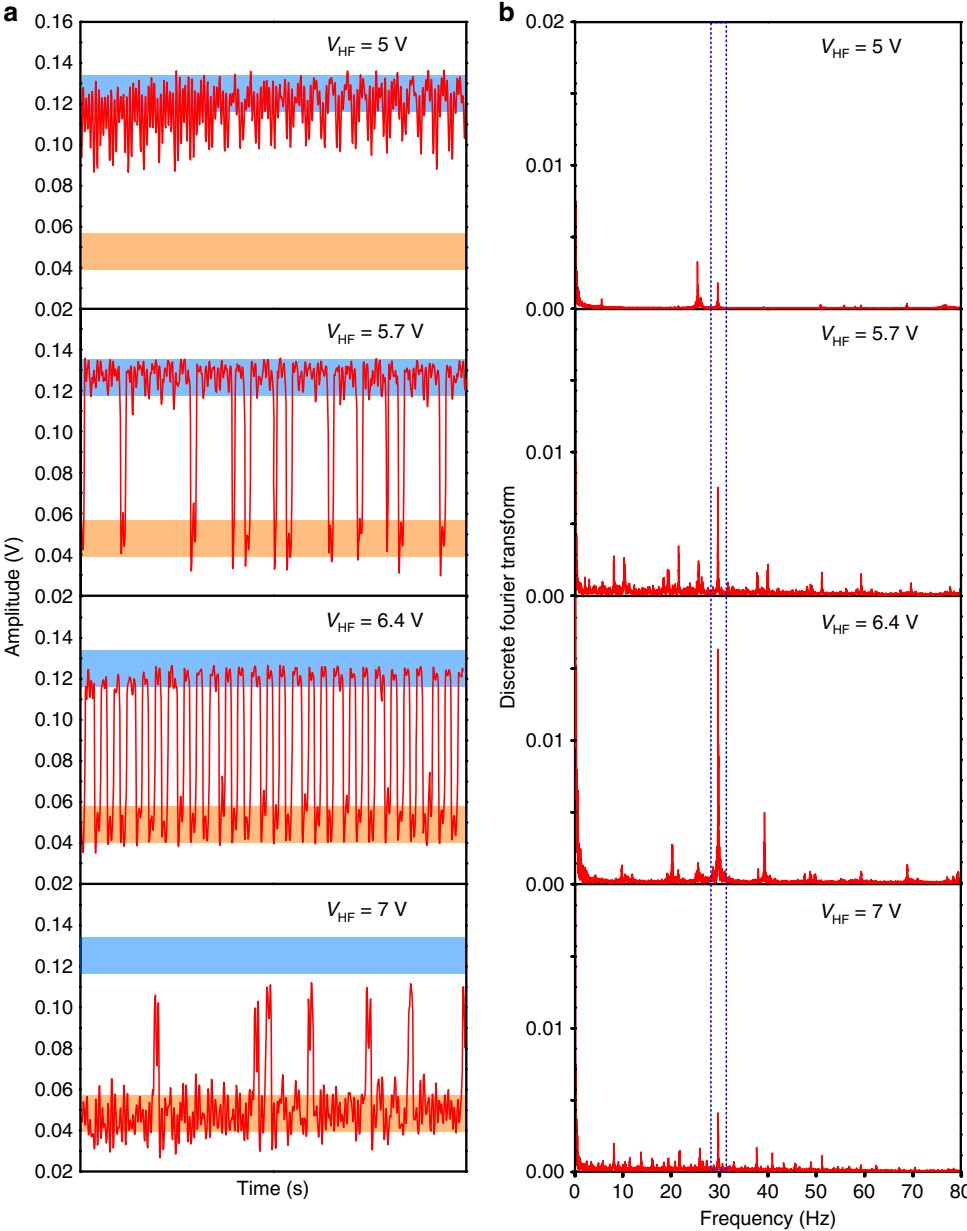

**Fig. 2 Time responses and Discret Fourier Transform as function of high-frequency modulation. a** Time series of the mechanical mode amplitude for a weak modulation with $\gamma = 0.1$, $\nu_m = 30$ Hz and for increasing high-frequency signal intensities $V_{HF} = 5V$, $5.7V$, $6.4V$ and $7.0V$ from top to bottom. The amplitudes of the two stable states at zero amplitude HF drive correspond to the blue and orange-shaded regions on each graph. **b** Discrete Fourier Transform of the time series displayed on the left. The vertical shaded lines enclose the modulation frequency $\nu_m$ of the weak signal.

inequalities: $\nu_m \ll \nu_{HF} \ll \nu_f$. The total applied voltage then writes:

$$V(t) = V_0[1 + \gamma \cos(2\pi\nu_m t) + \delta \cos(2\pi\nu_{HF}t)] \cos(2\pi\nu_f t) \quad (3)$$

Eq. (3) describes the total applied signal required to achieve amplification of the weak signal at $\nu_m$. The signals at $\nu_f$ and $\nu_{HF}$ can be externally controlled and triggered in order to respectively probe and enhance amplification of the weak signal which needs to be detected. Figure 2 shows time series of the mechanical motion amplitude for $\nu_m = 30$ Hz and increasing amplitudes of the amplitude modulation at high frequency $\nu_{HF} = 200$ kHz > $6500 \cdot \nu_m$. The system starts in its upper state (high amplitude state) where the small signal modulation is visible as a small intra-well motion. As the amplitude of the external driving increases, switching events between the two stable states become more prevalent. At first, occasional inter-well transitions occur, weakly locked to the modulation signal. For $V_{HF} = 6.4$ V, the system response gets completely synchronised with the applied weak and low-frequency modulation. Further increase of the additional external drive amplitude worsens the synchronisation and the system drops into its lower amplitude state, where a small intra-well modulation is visible. There is thus an optimal amplitude of the external drive which maximises the response amplitude. When the system is modulated close to the hysteresis turning points, it is more sensitive to noise induced fluctuations which are inherent in the experimental system, and this results in the observed aperiodic switchings in the weakly locked regions (cf. Fig. 2).

**Gain factor.** The gain or amplification factor can be inferred by quantifying the achieved spectral power amplification. For every time traces recorded on a time scale of 600 s, Discrete Fourier Transform (DFT) are performed. The resulting DFT spectra are presented in Fig. 2. They feature peaks, the most prominent being at the modulation frequency $\nu_m$. The achieved gain $M$ is then given by the ratio between the strength of the peak in the DFT spectrum at $\nu_m$ for a given amplitude of the external driving and its strength without external driving ($V_{HF} = 0$ V). The induced gain factor is presented in Fig. 3. The gain factor features a resonant-like behaviour: the gain factor first rapidly rises with the strength of the external driving, reaches a maximum for $V_{HF} = 6.4$ V and then drops. The maximum achieved gain factor is $M = 20$. Experimental noise modifies either the amplitude or the bistability region of the response. The probe at the frequency of the quasi-resonant forcing $\nu_f$ being fixed, this noise lead to fluctuations visible in the gain.

Vibrational resonance is governed by an amplitude condition. It occurs close to the transition from bistability to monostability, during which the effective potential of the slow variable evolves from a rapidly oscillating double well to a single well with a parametric dependence on the high-frequency signal amplitude and frequency[29]. As such, this phenomenon has some features in common with parametric amplification near the critical point.

**Theoretical analysis.** To figure out the origin of this resonant response, we introduce a simplified theoretical model and compare its results to our experimental findings. The original treatment of vibrational resonance in refs. [2,30] considers the motion of a nonlinear oscillator in a bistable potential, subject to a low-frequency signal and a high-frequency drive. Theoretical studies so far have mostly concentrated on studying the impact of the potential shape on the resonance[29,31,32], or the response to multi-frequency signals[33]. Interestingly, it was also noted in Gitterman[30] that one particularly important aspect of vibrational resonance was the ability to change the stability of some equilibria, or to have control over the shift of the resonance frequency. Our system only becomes a nonlinear oscillator if it is resonantly driven. Conversely, it cannot show a bistable response per se, whatever the sign of the stiffness parameter $\alpha$. However, with a quasi-resonant harmonic forcing, the nonlinear oscillator can become bistable. It is then interesting to examine in more details if an additional "high" frequency forcing can induce a resonance on a small amplitude signal.

The nanoelectromechanical system can be described in a good approximation as a forced nonlinear (cubic) Duffing oscillator[18]. Its dynamics can be modelled, in the limit of the small injection and the dissipation of energy by

$$\ddot{x} + \eta\dot{x} + \omega_0^2 x + \alpha x^3 = F[1 + \gamma \cos(\omega_m t) + \delta \cos(\Omega t)] \cos(\omega_f t),$$
$$(4)$$

where $x(t)$ accounts for the out-of-plane displacement of the membrane, $\eta$ is the effective damping, $\omega_0/2\pi$ is the natural oscillation frequency of the membrane, $\alpha$ is the nonlinear stiffness coefficient, $F$ is the amplitude of the modulated forcing with frequency $\omega_f/2\pi \equiv (\omega_0 + \Delta)/2\pi$, introducing the small detuning from resonance $\Delta$. The high-frequency amplitude modulation has an amplitude $F\delta$ and a frequency $\Omega/2\pi = \nu_{HF}$. The oscillation amplitude of the oscillator is the result of the beating of two frequencies: one fast at $\Omega$ and one slow at $\omega_m$. The parameters $\gamma$ and $\delta$ characterise the amplitude of the beating. By considering the following separation of timescales for the forcing frequencies $\omega_m \ll \Omega \ll \omega_0$, an amplitude equation for the time-averaged dynamics can be derived (see Methods). We start by deriving the equation for the amplitude of the forced nonlinear oscillator close to resonance ($\omega_f \sim \omega_0$) by looking for a solution in the form $x(t) = C(t)e^{i(\omega_0 + \Delta)t} + \text{cc}$ (where cc accounts for the complex conjugate term):

$$\partial_t C = -\frac{\eta}{2}C - i\Delta C + i\frac{3\alpha}{2\omega_0}|C|^2 C$$
$$- i\frac{F}{4\omega_0}(1 + \gamma \cos(\omega_m t) + \delta \cos(\Omega t)), \quad (5)$$

The strong timescale separation of the modulation frequencies motivates the introduction of a time-averaged variable $A$ over the short period $2\pi/\Omega$[34] such that

$$A(\tau) \equiv \frac{\Omega}{2\pi}\int_\tau^{\tau+2\pi/\Omega} C(t)dt.$$

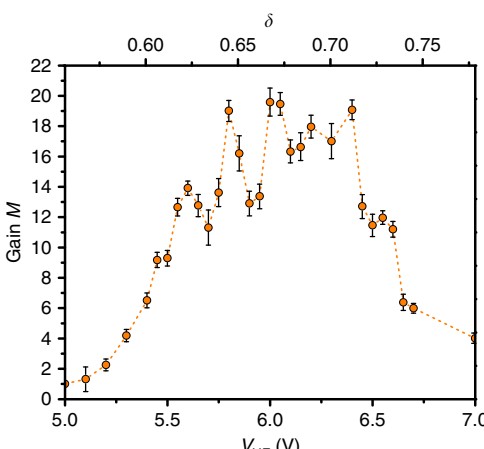

**Fig. 3 Amplification by vibrational resonance.** Gain factor $M$ with associated error bars as a function of the amplitude of the external driving.

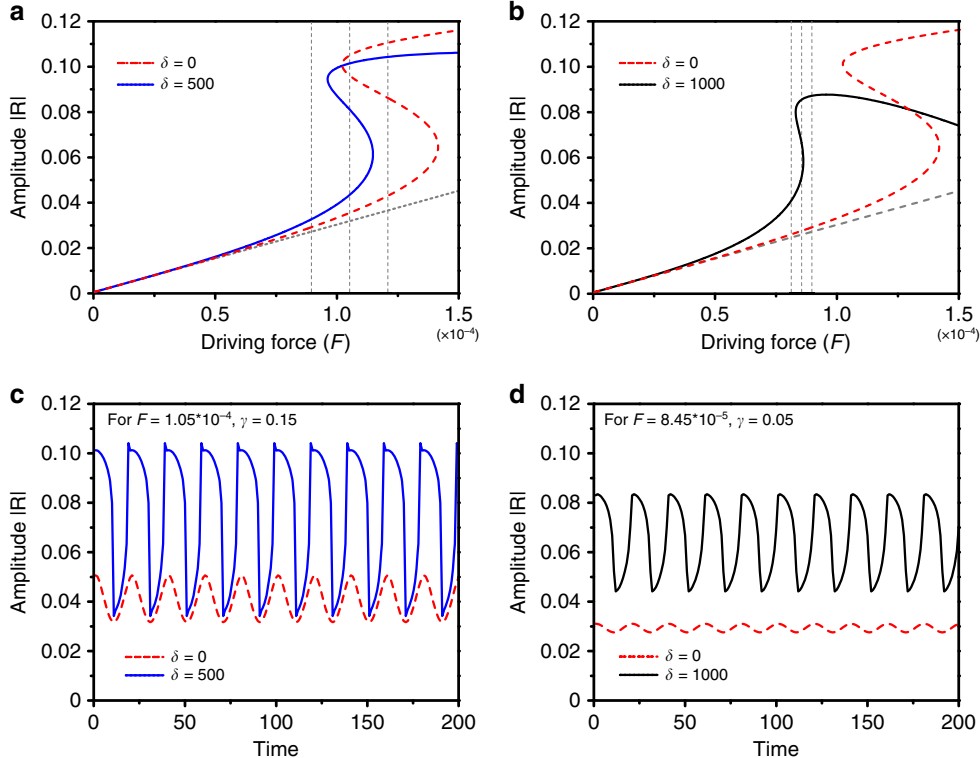

**Fig. 4 Numerical investigation of the steady-sate responses versus driving amplitude.** In **a**, **b**, the red dashed lines represents the steady state response |R| versus F for $\gamma = 0.15$ and $\delta = 0$ (no additive amplitude modulation). This is compared to **a** $\delta = 500$ (blue line), and **b** $\delta = 1000$ (black line). Vertical dashed lines in **a**, **b** lie at $F = F_0(1 \pm \gamma)$ and $F = F_0$, corresponding to the extreme and middle of the signal modulation. Other parameters are $\sigma = 0.0016$, $\eta = 0.001$, $\alpha = 0.4$. The grey lines mark the linear regime (small driving limit $F \to 0$). **c**, **d** : time traces of the response |R| corresponding to the maximum signal amplification in **a** and **b**, with respectively in **c** $F_0 = 1.05 \times 10^{-4}$ and $\delta = 0$ (red dashed line) or $\delta = 500$ (blue line) and in **d** $F_0 = 8.45 \times 10^{-5}$ and $\delta = 0$ (red dashed line) or $\delta = 1000$ (black line).

The amplitude equation for the averaged response writes

$$\partial_\tau A = -\frac{\eta}{2}A - i\left(\Delta - \frac{3\alpha F^2 \delta^2}{16}\right)A + i\frac{3\alpha}{2}|A|^2 A - i\frac{F}{4}(1 + \gamma\cos(\omega_m t)) \tag{6}$$

where we have introduced rescaled quantities: $\frac{F}{\omega_0} \to F$, $\frac{\delta}{\Omega} \to \delta$ and $\frac{\alpha}{\omega_0} \to \alpha$.

The averaged equation satisfies an amplitude equation with a renormalised detuning $\Delta - 3\alpha F^2 \delta^2/16$ which depends on the high-frequency driving amplitude. The most important aspect to note here is that the non-resonant and "high" frequency driving can modify the resonance behaviour of a nonlinear system. To study how the intermediate frequency $\Omega$ modifies the resonance region, we consider the polar representation $A = Re^{i\phi}/2$ with $\gamma = 0$, and solve for the steady state $\dot{R} = \dot{\phi} = 0$. We get the characteristic equation

$$\frac{\eta^2}{4}R^2 + \left[\left(\Delta - \frac{3\alpha F^2 \delta^2}{16}\right)R - \frac{3}{8}\alpha R^3\right]^2 = \frac{F^2}{4} \tag{7}$$

Note that in the limit of zero high-frequency amplitude modulation ($\delta \to 0$) we recover the deterministic forced Duffing resonator model. At this point, we highlight that the timescale separation hypothesis, $\omega_m \ll \Omega \ll \omega_0$, is central to obtain this result. Indeed, if we suppose $\omega_m \ll \omega_0 \ll \Omega$, i.e. a very high-frequency driving and average Eq. (4) before deriving the amplitude equation, then we cannot show evidence for vibrational resonance.

We numerically investigate the amplitude equation given by Eq. (7) for the parameters $\sigma = 0.0016$, $\eta = 0.001$, $\alpha = 0.4$ and

$\omega_m = 2\pi/200{,}000$. These parameters values are chosen to match the experimental ones. By fitting the nonlinear resonance curve (see Fig. 1b) we get the mechanical quality factor $Q_M = \eta^{-1}$ and the nonlinear spring term $\alpha$[18]. The signal modulation frequency is chosen to be much larger than mechanical quality factor to ensure almost adiabatic evolution. Figure 4 shows the steady-state response curves (inferred from Eq. (7)) versus the driving amplitude for different high-frequency amplitude modulation. Without any high-frequency drive ($\delta = 0$), the system displays a large hysteretic response (see Fig. 4a). A slow modulation of amplitude less than the hysteresis width would not produce any jump between the branches, hence would not produce any strong amplification of the signal at $\omega_m$. The addition of the high-frequency drive introduces an extra detuning which deforms the nonlinear response: in Fig. 4a, we observe that the centre of the hysteresis loop is shifted towards lower driving forces $F$, and that the width of the hysteresis shrinks as well. Since the signal modulation amplitude scales as $F\gamma$, this means that a smaller slow modulation amplitude $\gamma$ will be necessary to overcome the hysteresis width and produce large jumps between the lower and upper branch. This is the essence of the vibrational resonance phenomenon. In order to check this, we integrated numerically Eq. (7) with a slow amplitude signal at $\omega_m$. The results are shown on the time traces plotted in Fig. 4a, c. In the absence of high-frequency amplitude modulation ($\delta = 0$) the response is quasi linear since the system cannot jump between the lower and higher branches for the chosen modulation amplitude. The amplification factor is close to one if the system resides on the lower branch (same as the linear regime), or can be even smaller if it resides in the upper branch (de-amplification). When the driving $\delta$ is increased, the system is tuned into resonance and undergoes synchronous

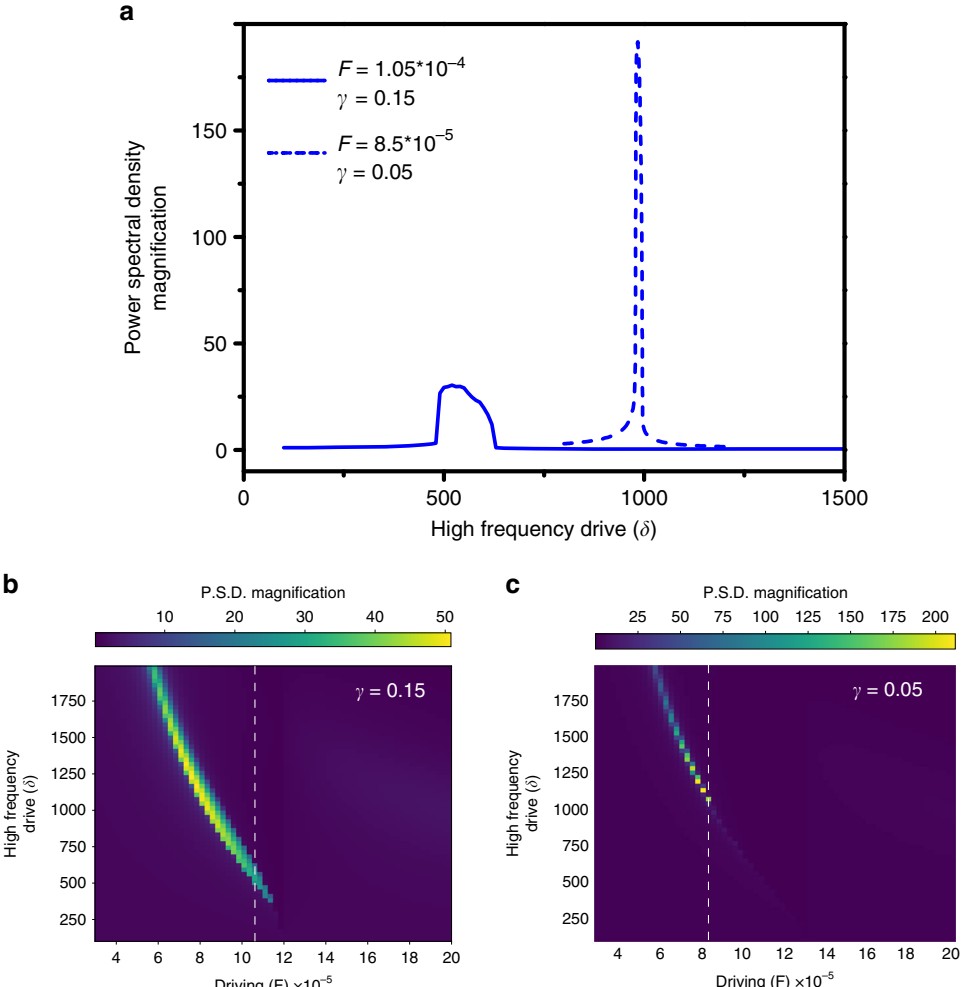

**Fig. 5 Simulated power spectral amplification for vibrational resonance. a** Power spectral density amplification $M = PSD(\omega, \delta)/PSD(\omega, 0)$ as a function of the high-frequency drive. Parameters for the solid blue line are : $\sigma = 0.0016$, $\eta = 0.001$, $\alpha = 0.4$, $\omega = 2\pi/200{,}000$, $F = 1.05 \times 10^{-4}$ and $\gamma = 0.15$. The parameters for the dashed blue line are the same except $F = 8.5 \times 10^{-5}$ and $\gamma = 0.05$ (lower resonant and low-frequency drivings). **b** Evolution of the power spectral density magnification as function of the forcing ($F$) and the detuning ($\delta$) for **b** $\gamma = 0.15$ and **c** $\gamma = 0.05$. White dashed vertical lines represent the cut at drivings used in **a**.

jumps with the signal driving frequency between the lower and upper branches (Fig. 4c). This corresponds to a large signal amplification, provided the signal amplitude is large enough, i.e. larger than the hysteresis width for the chosen parameters.

The amplification is shown in Fig. 5, rescaled to the response with zero high-frequency drive in the quasi linear case, i.e., on the lower branch. We plot both the amplitude ratio $M_a$ (ratio of the response amplitude with and without high-frequency drive) and the power-spectral density (PSD) ratio $M$. $M_a$ displays a sharp transition corresponding to the tuning of the system into the bistable region. When the high-frequency drive is not large enough, the system stays in the lower branch and the response is quasi-linear, leading to an amplification factor close to 1. When the bistable regime is reached, a large amplitude amplification is obtained. And for still higher $\delta$ the system stays in the upper branch where the response is sublinear, thus leading to de-amplification as expected. The same thing occurs in the PSD, except that the transition is less marked because the response of the system is highly nonlinear, hence the spectral energy is spread among the different harmonics. Note that we reach here, with the chosen parameters, a PSD amplification of the same order of magnitude as in the experiment. However, much larger amplification factors can be reached for other slightly different parameters, as illustrated in Fig. 5 where $M \sim 140$ is obtained for still

smaller linear driving signal not accessible in experiments. This important point is illustrated in Fig. 4b, d. If the signal strength is too small to overcome the hysteresis width, it is possible to increase the high-frequency drive to tune it into the resonance. As shown in Fig. 4b for $\delta = 1000$, a higher high-frequency modulation shifts the hysteresis curve further to the left, i.e., to lower overall forcing, but most importantly reduces the width of the hysteresis while not changing the hysteresis height too much. This makes it possible to amplify a much weaker signal by the vibrational resonance phenomenon. Note also that the amplification factor is even much larger in that case because of the already discussed different effect on the width and on the height of the hysteresis. This shows that it is necessary to tune both the high-frequency drive $\delta$ and the modulation strength $F$ to amplify optimally a signal of a given amplitude.

## Discussion

The previous analyses clearly indicate the primordial role of the high-frequency amplitude modulation and of the proper time-scale separation in such vibrational resonance phenomenon. The former allows to control the nonlinear resonance in order to amplify weak signals. The latter, while being compulsory for technical reasons in the theoretical analysis, could potentially be

relaxed in experiments. The exact value of the external drive frequency $\Omega$ is not critical at all, as long as it satisfies the timescale separation condition with $\omega_m \ll \omega_f$, $\Omega$ and if it remains non-resonant. Signal amplification results from the tuning into resonance of the non-linear response of the system. Amplification could occur also in the case of a non multivalued response, as long as the slope of the tuned response is large enough to ensure appropriate amplification. In principle, for any given signal amplitude, it is always possible to adjust both the forcing strength $F$ and the high-frequency modulation strength $\delta$ for vibrational resonance to occur. However, the maximum gain achievable will be a complicated function of all the system parameters. It will occur at the nascent bistability, i.e. when the hysteresis curve has an infinite tangent. Indeed, in this case, any small nonzero amplitude signal will be maximally amplified until saturation on the lower or upper branches. Ultimately, maximum gain achievable is limited by signal noise or noise in the system. Concerning the signal amplification frequency, its maximum is limited by the frequency response of the oscillator which is given by the damping rate (~1 kHz here). In order to push it further it is necessary to either decrease the mechanical quality factor of the nanomembrane or increase the resonant frequency with a similar quality factor.

By comparing the amplitude magnification curves in Figs. 3 and 5 we note a slight softening effect on the experimental gain curve whereas the theoretical one shows a sharp transition to high gain when the signal modulation is larger than the hysteresis curve turning points. This difference can be attributed to residual noise in the experiment which can modify the behaviour of the system close to the turning points of the nonlinear response.

At last, as observed in previous optical implementations of vibrational resonance[8,35] in non-parametrically forced bistable systems, the gain obtained seems higher than to the one observed in stochastic resonance for the same system. Even though vibrational resonance and stochastic resonance are based on different physical principles, this observation is verified in our system from a raw quantitative comparison with the stochastic resonance amplification[18].

In conclusion, we established and analysed the conditions for using vibrational resonance in order to enhance weak signals in a forced nonlinear oscillator, even if the system is initially monostable. The physical phenomenon is based on the resonance manipulation, thanks to a non-resonant, high-frequency amplitude-modulation drive obeying a timescale separation condition. We derived a model to describe vibrational resonance in a monostable, forced nonlinear oscillator which shows good agreement with our experimental results obtained on a forced nano-electromechanical membrane. This deterministic amplification method gives rise to high amplification factors, especially when compared to stochastic resonance[36]. As such, these results pave the way towards the design of novel architectures based on non-linear dynamic resonances for weak signal amplification, as currently done by quantum-limited Josephson parametric amplifiers[37] or, in the optical domain, by phase-sensitive amplifiers in the optical domain[38] to name a few. In a more general framework, it may open new avenues for the manipulation of non-linear resonances with the addition of a non-resonant driving field.

## Methods

**Fabrication of the InP resonator membrane**. The fabrication of the whole platform is based on a 3D heterogeneous integration process involving mainly four steps. First, a 400-nm-thick SiO$_2$ layer is deposited on the 260-nm-thick InP membrane, which is grown along with a 500-nm-thick InGaAs etch-stop layer on top of an InP (100) substrate by metal organic vapour phase epitaxy. Simultaneously, interdigitated electrodes (IDTs) arrays, displaying a finger period of 2 μm, finger length of 10 μm and electrode width of 500 nm, are deposited on a Si

substrate. The patterning process involves an electronic lithography, deposition of a 200-nm-thick gold layer and standard lift-off. The Si chip is then spin-coated with a 200-nm-thick DiVinylSiloxane-BenzoCycloButene (DVS-BCB) layer, thereby planarising the Si substrate. In the second step, the InP wafer is bonded on the Si substrate at high-temperature (300 °C) by positioning the SiO$_2$ layer atop the DVS-BCB layer and by using a vacuum wafer bonding technique[39]. The InP substrate and InGaAs etch-stop layer are then removed by chemical etching, leaving the residual 260-nm-thick InP membrane on the DVS-BCB-SiO$_2$ layer. In the third step, the InP membrane is patterned by standard e-beam lithography and dry-etching, to form a two-dimensional square-lattice photonic crystal of periodicity 532 nm, hole radius 181 nm and whole surface of $10 \times 20$ μm$^2$. It is clamped by four tethers of 2 μm length and 1 μm width, in order to reduce clamping losses. The alignment of the photonic crystal mirror with respect to the IDT's arrays, is performed with an accuracy better than 20 nm, by making use of alignment marks deposited beforehand on the Si substrate. Last, the photonic crystal membranes are released by under-etching the underlying 400-nm-thick SiO$_2$ layer, followed by a critical point drying step. The lateral InP suspension pads act as protective structures for the SiO$_2$ layer beneath them, leaving them anchored to the substrate.

**Measurement of the out-of-plane motion**. A He–Ne laser with wavelength of 633 nm is sent to the membrane. The reflectivity of the membrane is enhanced up to 50% by piercing a square lattice photonic crystal in it[40]. The laser is focused on the membrane via an objective with a NA of 0.4. The light reflected by the membrane is brought to interference with a strong local oscillator. A balanced homodyne detector locked on the drive frequency at the interferometer output is then used to decipher the amplitude and phase of the mechanical motion.

**Actuation of the mechanical oscillator**. The membrane is driven via the electrostatic force induced by the electrodes placed underneath. These electrodes are connected to an external signal generator which can go up to 50 MHz and is synchronised with a lock-in amplifier (HF2LI) which demodulates the detected signal at the actuation frequency. For vibrational resonance, the weak ($\nu_m$) modulated signal is generated by an another signal generator (Model Agilent 33522A) and combined with the additive ($\nu_{HF}$) signal in the HF2LI. This signal is then modulated at the frequency of the quasi-resonant forcing ($\nu_f$) and sent to the electrodes. The electrical signal obtained from the photodiodes (Thorlabs APD120A2) is time-recorded with the oscilloscope function of the HF2LI. The sampling frequency is 900 Hz, a much higher frequency than the one of the amplified signal.

**Derivation of the theoretical model**. Let us consider the timescale separation $\omega_m \ll \Omega \ll \omega_0$ and the resonance condition $\omega_f \sim \omega_0$. We first look for a solution to Eq. (4) using the ansatz $x(t) = C(t)e^{i(\omega_0 + \Delta)t} + cc$ (where $cc$ accounts for the complex conjugate term). After straightforward algebra, one gets an amplitude equation for the slow envelope $C(t)$, assuming that $\partial_{tt}C \ll \omega_0^2 C$ and $\partial_t C \ll \omega_0 C$:

$$\partial_t C = -\frac{\eta}{2}C - i\Delta C + i\frac{3\alpha}{2\omega_0}|C|^2 C - i\frac{F}{4\omega_0}(1 + \gamma\cos(\omega_m t) + \delta\cos(\Omega t)), \quad (8)$$

The amplitude equation (Eq. (8)) corresponds to the one of a forced oscillator with temporally modulated amplitude. Since we have a strong timescale separation of the modulation frequencies, $\omega_m \ll \Omega$, one can consider the averaged variable on the short period $2\pi/\Omega$

$$A(\tau) \equiv \frac{\Omega}{2\pi}\int_\tau^{\tau + 2\pi/\Omega} C(t)\,\mathrm{d}t$$

and homogenise the scales[41] by writing

$$C(t) = A(\tau) - \frac{\delta F}{4\omega_0 \Omega}e^{i\Omega t} \quad (9)$$

Considering that the envelope $A(\tau)$ is a slow variable ($\partial_\tau A \ll \omega_m A$), using the ansatz (9) in Eq. (8) and averaging over the period $2\pi/\Omega$, we get the amplitude equation for the averaged response

$$\partial_\tau A = -\frac{\eta}{2}A - i\left(\Delta - \frac{3\alpha F^2 \delta^2}{16\omega_0^3 \Omega^2}\right)A + i\frac{3\alpha}{2\omega_0}|A|^2 A - i\frac{F}{4\omega_0}(1 + \gamma\cos(\omega_m t)) \quad (10)$$

Introducing the notation $F' = \frac{F}{\omega_0}$, $\delta' = \frac{\delta}{\Omega}$ and $\alpha' = \frac{\alpha}{\omega_0}$ and omitting $'$, the equation reads

$$\partial_\tau A = -\frac{\eta}{2}A - i\left(\Delta - \frac{3\alpha F^2 \delta^2}{16}\right)A + i\frac{3\alpha}{2}|A|^2 A - i\frac{F}{4}(1 + \gamma\cos(\omega_m t)) \quad (11)$$

We further introduce a Madelung transform $A = Re^{i\phi}/2$ and $\gamma = 0$, and get

$$\dot{R} = -\frac{\eta}{2}R - \frac{F}{2}\sin(\phi) \quad (12)$$

$$R\dot{\phi} = -\left(\Delta - \frac{3\alpha F^2 \delta^2}{16}\right)R + \frac{3}{8}\alpha R^3 - \frac{F}{2}\cos(\phi) \quad (13)$$

At steady state, $\dot{R} = \dot{\phi} = 0$ and we finally get the characteristic equation

$$\frac{\eta^2}{4}R^2 + \left[\left(\Delta - \frac{3\alpha F^2\delta^2}{16}\right)R - \frac{3}{8}\alpha R^3\right]^2 = \frac{F^2}{4} \quad (14)$$

**High-frequency modulated forcing ($\Omega \gg \omega_o$).** Let us consider the nano-electromechanical system described by model Eq. (4) in the limit $\Omega \gg \omega_0 \sim \omega_f$. Due to the separation of temporal scales, one can analogously separate the temporal scales by means of the following change of variable

$$x = z(T) - \frac{F\delta\cos(\omega_f T)}{\Omega^2}\cos(\Omega t) \quad (15)$$

where the average over the rapid temporal variable reads

$$\langle x \rangle = \frac{\Omega}{2\pi}\int_T^{T+2\pi/\Omega} x(t)dt = z(T). \quad (16)$$

Introducing the ansatz (15) in Eq. (4) and taking the average over the rapid temporal variable, one gets after straightforward calculations,

$$\ddot{z} + \eta\dot{z} + \left(\omega_o^2 + \frac{3F^2\delta^2}{4\Omega^4}\right)z + \alpha z^3$$
$$= -\frac{3F^2\delta^2}{4\Omega^4}z\cos(2\omega_f t) + F[1 + \gamma\cos(\omega_m t)]\cos(\omega_f t). \quad (17)$$

Therefore, within this limit, the resonator natural frequency is modified to

$$\tilde{\omega}_0 = \sqrt{\omega_o^2 + \frac{3F^2\delta^2}{4\Omega^4}}. \quad (18)$$

In addition, the system has an extra parametric forcing term. Considering the forcing frequency close to the new resonant frequency $\omega_f = \tilde{\omega}_o + \tilde{\Delta}$ and looking for a solution of the form $z = ce^{i\omega_f t} + $ c.c. in Eq. (17), we obtain the amplitude equation for $C$

$$\partial_t C =; -\frac{\eta}{2}C - i\tilde{\Delta}C + i\frac{3\alpha}{2\tilde{\omega}_0}|C|^2C - i\frac{F}{4\tilde{\omega}_0}(1 + \gamma\cos(\omega_m t)) - \frac{3F^2\delta^2}{16\tilde{\omega}_o\Omega^4}\bar{C}, \quad (19)$$

By comparing Eq. (19) with Eq. (6), we notice that the terms are similar except for the complex conjugate term which is a signature of the additional parametric forcing. Therefore, we can conclude that the resonance of the system can be manipulated as in the previous case with the high-frequency forcing amplitude. However, the presence of a simultaneous parametric resonance does not allow to draw a simple conclusion on the amplification of the weak signal in that case, and we expect the parametric resonance to modify substantially the general physical picture.

**Manipulation of the resonance.** The characteristic equation Eq. (14) can be rearranged to yield a third order polynomial equation in $z = R^2$ such that

$$p(z) = \frac{9\alpha^2z^3}{64} + \frac{3}{64}\alpha z^2\left(3\alpha\delta^2F^2 - 16\Delta\right) + \frac{z}{256}\left(\left(3\alpha\delta^2F^2 - 16\Delta\right)^2 + 64\eta^2\right) - \frac{F^2}{4}$$

We look for extrema of the characteristic curve $p(z) = 0$ in the plane $(z, F)$ to compute the hysteresis width which yields two roots $z_\pm$

$$z_\pm = \frac{32\alpha\Delta - 6\alpha^2\delta^2F^2 \pm \sqrt{\alpha^2\left(\left(3\alpha\delta^2F^2 - 16\Delta\right)^2 - 192\eta^2\right)}}{18\alpha^2}$$

The hysteresis width can now be obtained by solving for $F_\pm$ in $p(z\pm) = 0$. One obtains the implicit formula

$$-27\alpha^3\delta^6F_\pm^6 \pm 9\alpha^2\delta^4F_\pm^4\left(\pm 48\Delta + \Delta_\pm\right)$$
$$-96\alpha F_\pm^2\left(6\left(\delta^2\left(4\Delta^2 + 3\eta^2\right) + 9\right) \pm \delta^2\Delta\Delta_\pm\right)$$
$$\pm 64\left(\pm 16\left(4\Delta^3 + 9\Delta\eta^2\right) + \left(4\Delta^2 - 3\eta^2\right)\Delta_\pm\right) = 0$$

with $\Delta_\pm = \sqrt{\left(3\alpha\delta^2F_\pm^2 - 16\Delta\right)^2 - 192\eta^2}$. An analytic expression for the hysteresis width $\Delta F = F_+ - F_-$ or the hysteresis centre $F_c = \frac{1}{2}\left(F_+ + F_-\right)$ cannot be expressed easily. However, for the set of parameters used in Fig. 4 we can compute both $\Delta F$ and $F_c$. With these parameters, the hysteresis width can be almost closed for a large high-frequency amplitude modulation $\delta$, whereas the hysteresis central frequency can be tuned in a large range, the same order of size as the original width.

## Data availability

All data and figures that support the findings of this study are available in Zenodo with the identifier https://doi.org/10.5281/zenodo.3595858.

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

## Acknowledgements

This work is supported by the French RENATECH network, the Marie Curie Innovative Training Networks (ITN) cQOM and the European Union's Horizon 2020 research and innovation programme under grant agreement No 732894 (FET Proactive HOT), and the Agence Nationale de Recherche projet ADOR (grant agreement no. ANR-19-CE24-0011-01). M.G.C. thanks the Millennium Institute for Research in Optics (MIRO) and FONDECYT projects Grants No. 1180903 for financial support.

## Author contributions

All authors planned the experiment and discussed the data. The sample was fabricated by A.C. and R.B., the measurement was carried out by A.C. in a setup build by A.C. and R.B. and A.C. analysed the data. M.G.C. and S.B. developed the theoretical model and I.R., M.G.C., S.B. and R.B. wrote the manuscript.

## Competing interests

The authors declare no competing interests.
