## [Peer Review File · Nature Communications]

Reviewers' Comments:

Reviewer #1:

Remarks to the Author:

The Authors present an experimental evidence of the phenomenon of vibrational resonance in nano-electro-mechanical nonlinear resonator. The peculiarity of the system at hand is the absence of inherent bistability which appears only in the case of the applied quasiperiodic driving. In these conditions the Authors observed experimentally the resonance-like behavior of the response at the low frequency depending on the amplitude of the additional high frequency signal which is typical for the phenomenon of vibrational resonance. Theoretical and numerical results also demonstrate the occurrence of vibrational resonance in the model which rather well describes the nano-electro-mechanical system. The results are novel and the paper rather clear written.

To my opinion the paper deserves publication. However, before publication some issues should be addressed.

1. What is the reason of the choice of so big frequency difference between low-frequency and high-frequency signals in the experiments? What is sampling frequency?
2. Scenario depicted in Fig. 2(a) (left) is slightly different from the usual one in vibrational resonance. For the HF amplitudes above resonance the response of the bistable system shows periodical switchings between two coexisting states with the frequency of the HF signal modulated slightly by LF signal. But here the system jumps to another steady state with no periodical switchings at high frequency. What is the reason? Is the same scenario observed in the numerical simulation?
3. The Authors write: "The shape of gain achieved in vibrational resonance consequently reflects the (here symmetrical) shape of the parametric gain in the system." From the figure 3 for the gain M one concludes that the shape for M is typical for the asymmetrical configuration of the bistable potential. In the symmetrical case the sharp transition to maximum gain is observed with increasing the amplitude HF signal. Is the same picture observed for M in the simulation as in Fig.3?
4. "Figure 5." The plot in the left panel is enough to characterize the factor gain M since the plot in the right panel is the same but in another scale.
5. " μ is the effective damping" . Please, correct. It should be " η "
6. "The electrical signal converted by the photodiodes...". Please, correct this phrase. It seems it should be something like that "The electrical signal obtained from the photodiode..."

Reviewer #2:

Remarks to the Author:

In the manuscript "Weak signal enhancement by non-linear resonance control in a forced nano-electromechanical resonator" Chowdhury et al. study the nonlinear response of a Duffing oscillator under off-resonant driving. In particular, they investigate a phenomena known as vibrational resonance, where a low frequency signal is enhanced under additional off resonant driving. Both the small signal and the off-resonant drive are a modulation of a near resonant force. The small signal is enhanced if its amplitude is similar to the width of the bistable region and the carrier driving strength falls within the bistable region.

The experimental work is an extension of their previous work (Ref. 14 PRL 2017) where they studied a related phenomena (stochastic resonance) in the same or a similar device.

The main result of the paper is that the off-resonant drive modifies the nonlinear response of the system. Specifically it changes the center and the width of the bistable region and therefore the conditions under which the small signal is enhanced. Potentially, this allows to amplify smaller signals since the bistable region becomes narrower.

In my opinion the paper is interesting and relevant to a very specialized audience interested in nonlinear resonance phenomena. However, it lacks the broader impact and context to justify publication in Nature Communications, and I suggest to submit to a more specialized journal, for

example Phys Rev E or Communications Physics.

- In terms of novelty, the broader scope of the field is missing. It is not clear what has been done before and what is the novelty of this paper. I suggest to emphasize this part in the introduction. To appeal to a wider audience, the authors should also relate their work to other systems which could exhibit similar phenomena. Duffing type resonators are quite ubiquitous, but a non-specialized reader would not know how general this dynamics is.

As it is now, the conditions under which signal amplifications occur seem to be contrived. Could the authors name an application, where a signal of the form Eq.3 is relevant? In the introduction, they mention inertial Brownian motion and GPS. However, it is not clear how those systems related to the one investigated in this paper.

- Regarding the discussion of the main result, the authors discuss how the off-resonant drive leads to a shift and narrowing of the bistable region on two specific examples. It seems that the discussion could be made more general by providing an analytical expression for the width and center of the bistable region, which can be extracted from Eq.7. From those, one could then formalize the vibrational resonance condition.

- The authors also make the point that the off-resonant driving frequency needs to be below the resonant drive frequency (timescale separation hypothesis). Can they provide an equivalent expression to Eq.7 for the case when $\Omega \gg \omega_0$, perhaps in the appendix?

- The gain is defined as follows:

"The achieved gain M is then given by the ratio between the strength of the peak in the DFT spectrum at ν_m for a given amplitude of the external driving and its strength without external driving"

However, this does not take into account that the drive not only increases the signal but also the noise. Hence, the gain should be defined as the ratio between the signal-to-noise ration with and without driving (see for example Ricci et al NATURE COMMUNICATIONS | DOI: 10.1038/ncomms15141).

- page 3 "Our system is a nonlinear oscillator with a small nonlinearity. It cannot show a bistable response per se, whatever the sign of the stiffness parameter". I find this sentence very confusing

- after Eq.4 μ is the effective damping. This should be η

- Fig.4 caption: kine => line

- page 4 "By fitting the nonlinear resonance curve we get " Is this fit shown in Fig.1b as dashed colored line? If so this is not very clear and a reference to the figure should be added to the text and the caption should mention that this is a fit.

- page 5 "As shown in Fig.4b ... higher high-frequency modulation shifts the hysteresis curve further to the right". right => left

- page 6 "At last, we can compare also the results of vibrational resonance with respect to those of stochastic resonance, ... suggest a higher gain factor with respect to the one observed for our system in stochastic resonance". This comparison is not clear at all. Where is the evidence that vibrational resonance provides higher gain than stochastic reference. Under which conditions can the two even be compared?

Reviewer #3:

Remarks to the Author:

Detailed comments are as follows:

1. Would strongly suggest the authors to compare their results with the current state of the art of weak signal amplifiers (with different other methods).
2. It seems that the amplification always comes with other modes (fig. 2). Any comments about how this affects the applications?
3. The gain has a huge fluctuation (in Fig. 3). Any comments?
4. More discussion about the limits of the gain coefficient? For example, how to push it further far beyond? And How to move for other frequency? What is the maximum frequency range?
5. It seems that the current setup is not really practical for applications. Can the authors elaborate more discussion about the perspectives about the potential applications, since the authors mentioned radio-frequency signal processing or sensing?

Rémy BRAIVE
Tel : 33 1 70 27 04 33
Mail : remy.braive@c2n.upsaclay.fr

Palaiseau, December 18th 2019

Dear Editor,

We have carefully considered the comments of Referees and have made some changes in the manuscript in response. The remarks of Referees have been extremely helpful in highlighting the points that need further clarification.

In what follows, we respond to the individual points raised by Referees highlighted in blue. Modifications are also highlighted in red in the article and Methods.

Reviewer #1:

The Authors present an experimental evidence of the phenomenon of vibrational resonance in nano-electro-mechanical nonlinear resonator. The peculiarity of the system at hand is the absence of inherent bistability which appears only in the case of the applied quasiperiodic driving. In these conditions the Authors observed experimentally the resonance-like behavior of the response at the low frequency depending on the amplitude of the additional high frequency signal which is typical for the phenomenon of vibrational resonance. Theoretical and numerical results also demonstrate the occurrence of vibrational resonance in the model which rather well describes the nano-electro-mechanical system. The results are novel and the paper rather clear written. To my opinion the paper deserves publication. However, before publication some issues should be addressed.

1. What is the reason of the choice of so big frequency difference between low-frequency and high-frequency signals in the experiments? What is sampling frequency?

One constrains concerns the relationship between the three frequencies coming into play in this configuration of vibrational resonance. To simplify theory and fit experiments, stringent inequalities have been used in this article. In order to remove the slow time scale and thus to simplify the theoretical discussion, a large frequency difference has to be used. Even if in the experiment, this is not mandatory (see below), frequencies have been well separated in order to be as close as possible to theory.

In the “Externally induced dynamics in the time domain” section, the following sentence “In this scenario reproduced in our experiment, the external drive takes the form of an additive amplitude modulation voltage of amplitude $V_{HF} \equiv \delta * V_0$ and frequency ν_{HF} such that $\nu_m \ll \nu_{HF} \ll \nu_f$ ”

has been modified in

“In this scenario reproduced in our experiment, the external drive takes the form of an additive amplitude modulation voltage of amplitude $V_{HF} \equiv \delta * V_0$ and frequency ν_{HF} . The criteria for enabling the

onset of vibrational resonance as predicted by theory (see section theoretical analysis), requires strong frequency inequalities : $\nu_m \ll \nu_{HF} \ll \nu_f$

Moreover, in the discussion section, we have already mentioned that amplification can be reached in the case of $\omega_m \ll \Omega, \omega_f$. To go further, one paragraph has been added in the Methods in order to discuss one restriction about frequencies. It highlights the fact that in this configuration, vibrational resonance with a driven nonlinear resonator, the resonance can also be manipulated even if the forcing modulation at ν_{HF} is higher than frequency of the driven resonator. However, amplification of the weak signal is not certain and needs to be further investigated.

Concerning the “sampling frequency”, if the referee meant the time resolution of the recording seen in Fig 2a, it is at about 1ms (or 900 Hz). A frequency much higher than the signal investigated here (30 Hz). Therefore, we are not missing any jumps between states. This precision has been added in the Methods “Actuation of the mechanical oscillator”:

“The sampling frequency is 900 Hz, a much higher frequency than the one of the amplified signal.”

If the referee meant the frequency at which the system is probed, it is at ν_f . This high frequency here corresponds to the near resonant drive that is needed to bring the resonator into the bistable regime. The low frequency is the signal frequency. Its frequency range necessarily has to be lower than the resonant drive. The maximum signal frequency usable in this system corresponds to the adiabatic response of the system and will therefore be limited by the damping. Here we then expect that the maximum signal frequency will be of the order of $1/\eta$ (~1kHz here). In order to push it further it is necessary to increase the mechanical quality factor. However, it should be noted that the phenomenon will persist (though, possibly much less pronounced) if the signal frequency is out of the previous range.

2. Scenario depicted in Fig. 2(a) (left) is slightly different from the usual one in vibrational resonance. For the HF amplitudes above resonance the response of the bistable system shows periodical switchings between two coexisting states with the frequency of the HF signal modulated slightly by LF signal. But here the system jumps to another steady state with no periodical switchings at high frequency. What is the reason? Is the same scenario observed in the numerical simulation?

Indeed, we see in Fig. 2(a) (left) bottom that the switches at high HF drive amplitude don't reach the expected upper branch obtained at lower HF drive amplitude. This is a signature of the hysteresis deformation induced by the HF drive amplitude: as can be seen on Figure 4a or 4b, when the HF amplitude is higher than the upper branch has a tendency to have a lower amplitude (while the lower branch remains at the same amplitude). This is not taken into account in Figure 2 where the blue and orange bands are just guides for the eye and are plotted for low HF drive amplitude. Concerning the aperiodic switches, the origin is attributed to inherent residual noise present in the experiment and which induces the aperiodic switches. They are not reproduced by our deterministic model. However, we mention that in principle, for a larger signal modulation frequency and amplitude, chaotic switching is expected.

In Figure 2a caption we modify: The amplitude of the two stable states corresponds to the blue and orange-shaded regions on each graph.

replaced by

The amplitudes of the two stable states at zero amplitude HF drive correspond to the blue and orange-shaded regions on each graph.

In page 2 before “gain factor” section, we add

“When the system is modulated close to the hysteresis turning points, it is more sensitive to noise induced fluctuations which are inherent in the experimental system, and this results in the observed aperiodic switchings in the weakly locked regions (cf. figure 2).”

3. The Authors write: “The shape of gain achieved in vibrational resonance consequently reflects the (here symmetrical) shape of the parametric gain in the system.” From the figure 3 for the gain M one concludes that the shape for M is typical for the asymmetrical configuration of the bistable potential. In the symmetrical case the sharp transition to maximum gain is observed with increasing the amplitude HF signal. Is the same picture observed for M in the simulation as in Fig.3?

The sentence about gain symmetry brings confusion here. Indeed, the parametric gain has not to be symmetric, strictly speaking, only for certain parameters. Therefore we remove the sentence.

4. “Figure 5.” The plot in the left panel is enough to characterize the factor gain M since the plot in the right panel is the same but in another scale.

This is true that left and right panels in Fig. 5 are with the same amplitude of the modulated forcing (F) and amplitude of the fast frequency (δ). The difference between the two graphs is the feature plotted in the y-axis; either the amplitude amplification (left) or the power spectral density magnification (right).

For experimental results (Fig. 3), only the power spectral density magnification is shown. In order to be compared to the experimental data in Fig. 4, we only keep the right panel. This highlights that the amplification reached in experiment and theory are of the same order of magnitude for reasonable parameters.

The caption of figure 5 has been modified as follow:

“Power spectral density amplification $M = \text{PSD}(\omega, \delta) / \text{PSD}(\omega, 0)$ as a function of the high frequency drive. Parameters for the solid blue line are: $\sigma = 0.0016$, $\eta = 0.001$, $\alpha = 0.4$, $\omega = 2\pi / 200000$, $F = 1.05 \cdot 10^{-4}$ and $\gamma = 0.15$. The parameters for the dashed red and blue lines are the same except $F = 8.5 \cdot 10^{-5}$ and $\gamma = 0.05$ (lower resonant and low frequency drivings).”

5. “ μ is the effective damping” . Please, correct. It should be “ η ”

This has been corrected

6. “The electrical signal converted by the photodiodes...”. Please, correct this phrase. It seems it should be something like that “The electrical signal obtained from the photodiode...”

This has been corrected

Reviewer #2:

In the manuscript "Weak signal enhancement by non-linear resonance control in a forced nano-electromechanical resonator" Chowdhury et al. study the nonlinear response of a Duffing oscillator under off-resonant driving. In particular, they investigate a phenomena known as vibrational resonance, where a low frequency signal is enhanced under additional off resonant driving. Both the small signal and the off-resonant drive are a modulation of a near resonant force. The small signal is enhanced if its amplitude is similar to the width of the bistable region and the carrier driving strength falls within the bistable region.

The experimental work is an extension of their previous work (Ref. 14 PRL 2017) where they studied a related phenomena (stochastic resonance) in the same or a similar device. The main result of the paper is that the off-resonant drive modifies the nonlinear response of the system. Specifically it changes the center and the width of the bistable region and therefore the conditions under which the small signal is enhanced. Potentially, this allows to amplify smaller signals since the bistable region becomes narrower.

In my opinion the paper is interesting and relevant to a very specialized audience interested in nonlinear resonance phenomena. However, it lacks the broader impact and context to justify publication in Nature Communications, and I suggest to submit to a more specialized journal, for example Phys Rev E or Communications Physics.

- In terms of novelty, the broader scope of the field is missing. It is not clear what has been done before and what is the novelty of this paper. I suggest to emphasize this part in the introduction. To appeal to a wider audience, the authors should also relate their work to other systems which could exhibit similar phenomena. Duffing type resonators are quite ubiquitous, but a non-specialized reader would not know how general this dynamics is.

As it is now, the conditions under which signal amplifications occur seem to be contrived. Could the authors name an application, where a signal of the form Eq.3 is relevant? In the introduction, they mention inertial Brownian motion and GPS. However, it is not clear how those systems related to the one investigated in this paper.

As mentioned by Referee 2, Duffing type resonators are quite ubiquitous. It has been successfully used to model a variety of physical processes such as stiffening springs, beam buckling, nonlinear electronic circuits, superconducting Josephson parametric amplifiers, and ionization waves in plasmas to name a few. Moreover, Duffing model is an example of dynamical system that may exhibit chaotic behaviour.

In order to emphasize this, the following sentence has been added in the introduction section:

" This phenomenon can be fully understood thanks to the ubiquitous Duffing model which, beyond nanomechanics, can be used for superconducting Josephson amplifier [Vijay], ionization waves in plasma [Nambu, Roy-Layinde] to describe complex spatiotemporal behaviours such as chimera states [Clerc]. In the frame of well-known Duffing model, the oscillator then features two equilibrium states..."

Equation 3 reveals the expression of the voltage seen by the resonator. In no case, it is the signal to be amplified in any kind of potential application. The amplified signal is the one with frequency ν_m . The two other frequencies could be seen as tools for probing the amplification (mechanical frequency of the resonator, ν_f) and supporting the amplification (ν_{HF}).

In the case of applications, the weak signal collected by any means (optically, electrically ...) could then be combined via mixers and bias-T to the two known voltages (frequencies and amplitudes) in order to amplify the weak signal.

Main text has been changed as follow in order to avoid this misunderstanding:

“Eq (3) describes the total applied signal required to achieve amplification of the weak signal at v_m . The signals at v_f and v_{HF} can be externally controlled and triggered in order to respectively probe and enhance amplification of the weak signal which needs to be detected.”

Concerning the introduction where GPS and inertial Brownian motion is mentioned, this part has been completely rewritten. The sentence that the referee refers to is now:

“We argue that this effect could be used besides the one we are presenting here, as a general mechanism for nonlinear resonance manipulation. Beyond these fundamental aspects, potential applications encompass various fields such as telecommunications, where the data are encoded on a low-frequency modulation applied to a high-frequency carrier, or, more prospectively, microwave signals processing and sensing such as accelerometers featuring enhanced sensitivities via optomechanically-driven signal amplification (Krause et al, Nature Photon. 6, 768 (2012)). Bichromatic signals are in addition pervasive in many other fields including brain-inspired architecture mimicking neural networks (Qin et al, Cognitive Neurodynamics 12, 509 (2018)) where bursting neurons may exhibit two widely different time scales.”

- Regarding the discussion of the main result, the authors discuss how the off-resonant drive leads to a shift and narrowing of the bistable region on two specific examples. It seems that the discussion could be made more general by providing an analytical expression for the width and center of the bistable region, which can be extracted from Eq.7. From those, one could then formalize the vibrational resonance condition.

By rewriting Eq. 7 and looking at its extrema, we were able to express an implicit formula. However, no analytical expression can be expressed from this latter one. As a consequence, expression of the width and the centre of the bistable region cannot be addressed in a general framework. This is derived in details in the appendix untitled “Manipulation of the resonance”.

For the set of parameters used in this article, we can see in the figure below (not added in the manuscript) that the hysteresis central frequency can be tuned in a large range, of the same order of size as the original width (left), whereas the hysteresis width can be almost closed for a large high-frequency amplitude modulation δ (right).

Figure 1: Hysteresis manipulation with the high-frequency modulation amplitude δ . (Left) Evolution of the hysteresis centre F_c and (right) the hysteresis width ΔF . The parameters are the same as in Figure 5.

- The authors also make the point that the off-resonant driving frequency needs to be below the resonant drive frequency (timescale separation hypothesis). Can they provide an equivalent expression to Eq.7 for the case when $\Omega \gg \omega_0$, perhaps in the appendix?

In this limit, the system simultaneously exhibits a normal and a parametric resonance at a manipulable effective frequency. In order to highlight this effect, a paragraph "High frequency modulated forcing ($\Omega \gg \omega_0$)" has been added in the appendix as suggested by the referee.

For the sake of readability, the paragraph is not reproduced here.

- The gain is defined as follows:

"The achieved gain M is then given by the ratio between the strength of the peak in the DFT spectrum at ν_m for a given amplitude of the external driving and its strength without external driving" However, this does not take into account that the drive not only increases the signal but also the noise. Hence, the gain should be defined as the ratio between the signal-to-noise ration with and without driving (see for example Ricci et al NATURE COMMUNICATIONS | DOI: 10.1038/ncomms15141).

In the case of the article given as an example by the referee, amplification is reached thanks to added noise in the system. This is the so-called stochastic resonance. As a consequence, the signal to noise ratio is indeed an important feature. However, in the case discussed in our manuscript, noise is replaced by an additional modulation whose amplitude is the key element for amplification. Thus, any noise injected in our system (electrical noise, external mechanical vibrations...) should be averaged over one recording (60 seconds long). Furthermore, we could expect that it is the same for every recording.

In the figure 2 (below), we show the evolution of the noise floor of the Discrete Fourier Transform (Fig 2 right of the manuscript) as function as the amplitude of the driving. Horizontal lines are guide for the eyes in order to evidence the mean value and the standard deviation ($\pm\sigma$ in blue and $\pm 2\sigma$ in light blue). We see that the noise is not relevant for these measurements.

Figure 2 : Evolution of the noise floor in D.F.T. spectrum as function of the driving voltage. As a guide for the eye, the mean value (in red), the variance at $\pm\sigma$ (dashed blue line) and at $\pm 2\sigma$ (dotted light blue line) are shown.

- page 3 "Our system is a nonlinear oscillator with a small nonlinearity. It cannot show a bistable response per se, whatever the sign of the stiffness parameter". I find this sentence very confusing

This sentence has been rephrased as followed:

"Our system becomes a nonlinear oscillator if it is resonantly driven. Conversely, it cannot show a bistable response per se, whatever the sign of the stiffness parameter α ."

- after Eq.4 μ is the effective damping. This should be η : This has been changed.

- Fig.4 caption: kine => line : This has been changed.

- page 4 "By fitting the nonlinear resonance curve we get " Is this fit shown in Fig.1b as dashed colored line? If so this is not very clear and a reference to the figure should be added to the text and the caption should mention that this is a fit.

It is true that there was no fitting in Fig 1b. Fig 1 has been changed by adding the fit to the nonlinear amplitude response of the mechanical mode for the sweep up. The caption has been changed accordingly. ("... $V_0=9V$. **A theoretical fitting of the forced amplitude response curve (sweep up) is shown (black line).** The vertical ...)

- page 5 "As shown in Fig.4b ... higher high-frequency modulation shifts the hysteresis curve further to the right". right => left

This has been changed.

- page 6 "At last, we can compare also the results of vibrational resonance with respect to those of stochastic resonance, ... suggest a higher gain factor with respect to the one observed for our system in stochastic resonance". This comparison is not clear at all. Where is the evidence that vibrational resonance provides higher gain than stochastic reference. Under which conditions can the two even be compared?

For a similar structure, authors have already demonstrated stochastic resonance of a signal at 50 Hz.

In this configuration, the gain factor was, at maximum, a factor 6 experimentally (and a factor 10 theoretically).

From a strictly quantitative point of view, a direct comparison of the two results shows a better amplification in the case of vibrational resonance. From an experimental point of view, the required conditions to fulfil these processes are much easier to achieve in the case of vibrational resonance. No time matching condition is required. Beyond these comments, the physics behind these two processes is quite different.

It is not our intention in this article to have a definitive statement about the most efficient process for small signal amplification based on bistable resonator.

The sentence has been rephrased as followed in order to avoid this confusion:

At last, as observed in previous optical implementations of vibrational resonance [8, 31] in non-parametrically forced bistable systems, the gain obtained seems higher than to the one observed in stochastic resonance for the same system. Even though vibrational resonance and stochastic resonance are based on different physical principles, this observation is verified in our system from a raw quantitative comparison with the stochastic resonance amplification [18]."

Reviewer #3 :

Detailed comments are as follows:

1. Would strongly suggest the authors to compare their results with the current state of the art of weak signal amplifiers (with different other methods).

Before anything else, the scope of the results here, is not yet to propose a novel architecture for weak signal amplification. It concerns a deep investigation of the criteria enabling the onset of non-linear resonances in a forced monostable system. Establishing the potential of such effects for building amplifiers is beyond the scope of our manuscript.

This being said, it is true that different other methods are available to amplify small signal, based on different kinds of physics and thus different devices; from electronic amplifiers to optical amplifiers. In the following, we discuss these different devices, their uses and limitations, and compare to our structure.

Electrical amplifiers are based on nonlinear elements such as transistors. Typically, such components exhibit a change in their dynamics thanks to a third terminal namely a gate e.g. Junction Field Effect Transistor (JFET) or Bipolar Junction Transistor (BJT). In these devices, amplification is generally limited by distortion.

Still in the electrical domain, a number of different circuit elements (superconducting island or loop...) can be used to couple an external signal (electrical charge or magnetic flux) to the properties of the junction (critical current in the junction). Small variations in this current can significantly change the transport properties of the junction under appropriate bias condition, giving rise to a sensitive amplifier with near quantum limited noise performance (Zorin, Phys. Rev. Lett. 76, 4408 (1996), Schoelkopf et al, Science 280,1238 (1988)). Josephson junction-based device such as Josephson Bifurcation Amplifier (JBA) or Cavity Josephson Bifurcation Amplifier (CBA) exploit the anharmonicity of the superconducting junction oscillator for enhanced measurement sensitivity.

On this basis, the structure developed in the manuscript is based on the same principle. Two main experimental differences could be highlighted: the environmental conditions at which structure are working and their respective working frequency. Oscillator used in JBA are based on superconducting junction working in a dilution fridge in the GHz range, but our structure using a mechanical resonator is working in a less harsh environment (ambient temperature; low pressure), in the MHz range. By working at higher frequencies, JBA or related device have a much larger bandwidth and above all, achieved quantum limited amplification.

From the optical point of view, there is the well-known and commercialized Erbium-Doped Fiber Amplifier (EDFA) in which the output power can reach typical values between 20 to 40 dBm in the C-band for infrared optical communications for a minimum input as low as -20 dBm (10 μ W). This 10 μ W in C-band can easily be read by commercial photodetectors. Thus, to the best of our knowledge, EDFA are not used to amplify unreadable signals but rather to achieve high power in order to get access to new effects such as nonlinear optics in fibers or nanostructures, e.g. frequency combs (Kippenberg et al, Science 332, 555 (2011)) .

Each of these devices is amplifying the signal in the same domain from electronics to electronics or optics to optics. Even though the achieved experimental amplification shown in the manuscript does not reach the level of other amplifying devices, the main difference here is that we amplify electrical weak signals in the optical domain through a nonlinear mechanical device. With the added panel in

Figure 5, we see that our current measurements were not at the optimum drivings (High frequency driving and F). By properly adjusting them, we could reach an amplification of about 200 in amplitude.

In the discussion section, to better emphasize the positioning of our manuscript, the already existing technologies and put our work in a broader context, we have added the following sentence:

“As such, these results pave the way towards the design of novel architectures based on non-linear dynamic resonances for weak signal amplification, as currently done by quantum-limited Josephson parametric amplifiers [Macklin et al] or, in the optical domain, by phase-sensitive amplifiers in the optical domain [Tong et al] to name a few.”

2. It seems that the amplification always comes with other modes (fig. 2). Any comments about how this affects the applications?

It is true that spurious peaks appear in the DFT spectrum, whose presence is highly variable with the high voltage driving (V_{HF}). In applications, these peaks could be easily eliminated by frequency filtering or by averaging.

3. The gain has a huge fluctuation (in Fig. 3). Any comments?

The experiment is not insensitive to external and technical noise such as mechanical vibrations or electrical noise. Any of these noises slightly modify either the position of the drive in the bistability region or the amplitude of the mechanical response. As such, inevitable experimental noise modify / blur the amplification from one value to another. This can already be seen on time traces of Fig. 2. At $V_{HF} = 6.4V$, the maximum of the time trace is a little bit lower than 1 sec later. Over one-minute time frame, amplitude fluctuations are visible.

The following sentence has been added in the “Gain factor” section:

“Experimental noise modifies either the amplitude or the bistability region of the response. The probe at the frequency of the quasi-resonant forcing ν_f being fixed, this noise lead to fluctuations visible in the gain.”

4. More discussion about the limits of the gain coefficient? For example, how to push it further far beyond? And How to move for other frequency? What is the maximum frequency range?

We provide additional numerical simulations on the gain attainable. This can be seen in Figure 5 where one of the panels has been removed and we added two color maps of the gain versus two driving terms (F and δ). Maximum gain will be obtained when the hysteresis curve has an infinite tangent, i.e. in the nascent bistability case. In that case, any small non zero amplitude signal will be maximally amplified until saturation on the lower and upper branches. Maximum gain will be ultimately limited by noise.

The maximum frequency range corresponds to the adiabatic response of the system and will therefore be limited by the damping of the system. Here we then expect that the maximum signal frequency will be of the order of $1/\eta$ ($\sim 1kHz$ here). In order to push it further it is necessary to increase the mechanical quality factor. However, it should be noted that the phenomenon will persist (though, possibly much less pronounced) if the signal frequency is out of the previous range

Since the limitation is basically related to intrinsic parameters of the resonator under study, one may think either increasing the Q-factor with similar structures, having another resonator vibrating at higher frequency or even both. In the case of the structure used in this manuscript, an out-of-plane displacement of the membrane, few MHz frequencies can be achieved but not much than this and the damping rate is usually of the order of a few tens of kHz. To gain several orders of magnitude in frequency, one may think to optomechanical crystals (Eichenfield et al, Nature 462, 78 (2009)) whose Q-factor is about few thousands but at much higher frequencies (few GHz) or even hundreds of GHz (Esmann et al, Optica 6, vol. 7, p 854 (2019)) .

We add in the discussion section

“It will occur at the nascent bistability, i.e. when the hysteresis curve has an infinity tangent. Indeed, in this case, any small nonzero amplitude signal will be maximally amplified until saturation on the lower or upper branches. Ultimately, maximum gain achievable is limited by signal noise or noise in the system. Concerning the signal amplification frequency, its maximum is limited by the frequency response of the oscillator which is given by the damping rate (~1 kHz here). In order to push it further it is necessary to increase the mechanical quality factor of the nanomembrane or increase the resonant frequency with a similar quality factor.”

5. It seems that the current setup is not really practical for applications. Can the authors elaborate more discussion about the perspectives about the potential applications, since the authors mentioned radio-frequency signal processing or sensing?

At present, this device just demonstrates a proof of principle of weak signal amplification in a mechanical nanoresonator thanks to nonlinear resonance manipulation. However, without foreshadowing the future, these devices or more realistically the physical principle might be the first building block towards a functional device as Josephson Bifurcation Amplifier are nowadays. In this frame, we were envisioning two potential applications such as microwave signal and sensing.

For the microwave part, one might think about the so-called optomechanical crystal as a potential application for low phase noise X-band microwave oscillators (Metcalfe, Applied Physics Reviews 1, 03105 (2014)). Over the last decade, photonic crystal cavities have demonstrated their ability of hosting mechanical modes colocalized with their optical counterpart. This so called optomechanical crystal mechanically resonates at a frequency of a few GHz (Eichenfield et al, Nature 462, 78 (2009)). Mechanical nonlinearity used in this manuscript in the MHz could eventually be demonstrated in the GHz range, thus, opening new avenues in amplifying weak signals at much higher frequency that we show here with mechanical systems in the microwave domain.

Similarly, accelerometers have been demonstrated using optomechanical (Krause et al, Nature Photon. 6, 768 (2012)). As such, vibrational resonance could be used to amplify the AC acceleration evidenced in this article. They are using a shake table in order to create the physical displacement. It could be seen as our low frequency signal which need to be amplified.

The last sentence of the abstract has been rephrased as follow:

“Our results illustrate a general mechanism which might have applications in the fields of microwave signal amplification or sensing for instance.”

The last sentence of the introduction:

“We argue that this effect could be used besides the one we are presenting here, for another purpose using pervasive biharmonic signals such as inertial Brownian motors [MACHURA2010445] and the Global Positioning System [Kaplan05], and illustrates thus a general mechanism for nonlinear resonance manipulation.”

has also been changed to

“We argue that this effect could be used besides the one we are presenting here, as a general mechanism for nonlinear resonance manipulation. Beyond these fundamental aspects, potential applications encompass various fields such as telecommunications, where the data are encoded on a low-frequency modulation applied to a high-frequency carrier, or, more prospectively, microwave signals processing and sensing such as accelerometers featuring enhanced sensitivities via optomechanically-driven signal amplification (Krause et al, Nature Photon. 6, 768 (2012)). Bichromatic signals are in addition pervasive In many other fields including brain-inspired architecture mimicking neural networks (Qin et al, Cognitive Neurodynamics 12, 509 (2018)) where bursting neurons may exhibit two widely different time scales.”

Reviewers' Comments:

Reviewer #1:

Remarks to the Author:

In revised version of the paper the Authors have taken into account suggestions and comments in a satisfactory way. Therefore, I recommend the paper for publication.

Reviewer #2:

Remarks to the Author:

The authors addressed all my previous concerns.

potential typos:

p.6 What is "innity tangent"? Should this be infinity?

p.6 "Concerning the signal amplification frequency, its maximum is limited by the frequency response of the oscillator which is given by the damping rate (~1 kHz here). In order to push it further it is necessary to either increase the mechanical quality factor .. "

Increasing the mechanical quality factor reduces the bandwidth. Doesn't this also reduce the maximum frequency? Should the previous sentence read "decrease the mechanical quality factor"?

Reviewer #3:

Remarks to the Author:

The authors have addressed all my technical concerns. However, the responses present a new concern: whether the manuscript is interesting for general audience. This was also the concern of reviewer 2, which has not been fully addressed this round. Would suggest the authors to emphasis the key novelties of the manuscript.

Rémy BRAIVE
Tel : 33 1 70 27 04 33
Mail : remy.braive@c2n.upsaclay.fr

Palaiseau, February 17th 2020

We have carefully considered the comments of Referees and have made some changes in the manuscript in response. In what follows, we respond to the individual points raised by Referees highlighted in blue text. Modifications are also highlighted in red in the article and Methods.

Reviewer #1 (Remarks to the Author):

In the revised version of the paper the Authors have taken into account suggestions and comments in a satisfactory way. Therefore, I recommend the paper for publication.

Reviewer #2 (Remarks to the Author):

The authors addressed all my previous concerns.

potential typos:

p.6 What is "innity tangent"? Should this be infinity?
This typo has been corrected.

p.6 "Concerning the signal amplification frequency, its maximum is limited by the frequency response of the oscillator which is given by the damping rate (~ 1 kHz here). In order to push it further it is necessary to either increase the mechanical quality factor .. "
Increasing the mechanical quality factor reduces the bandwidth. Doesn't this also reduce the maximum frequency? Should the previous sentence read "decrease the mechanical quality factor"?

This is a mistake. Referee #2 is right. By increasing the quality factor, the maximum frequency at which the system will amplify the signal, will be lower. As such, we need to decrease the mechanical quality factor towards higher frequencies. This was an error which has been changed.

Reviewer #3 (Remarks to the Author):

The authors have addressed all my technical concerns. However, the responses present a new concern: whether the manuscript is interesting for general audience. This was also the concern of reviewer 2, which has not been fully addressed this round. Would suggest the authors to emphasis the key novelties of the manuscript.

During the first stage of reviewing, the potential interest for a general audience has indeed already been addressed by Reviewer #2. Our previous answer tackled this concern at various levels, from the point of view of the theory to the potential applications of this phenomenon, and seemed to fulfil the expectations of reviewer #2.

The model developed here in order to understand the occurrence of vibrational resonance in a driven resonator can be applied to many different systems. This has been emphasized in the previous version with the following sentence:

« ... This phenomenon can be fully understood thanks to the ubiquitous Duffing model which, beyond nanomechanics, can be used for superconducting Josephson amplifier [11], ionization waves in plasma [12, 13] to describe complex spatiotemporal behaviors such as chimera states [14]. In the frame of the well-known Duffing model ... »

Furthermore, some potential applications of vibrational resonances in various fields (nanoelectronics, optomechanics, neuromimeticism ...), have also been described in the introduction and conclusion of the previous version and copied here:

“...We argue that this effect could be used besides the one we are presenting here, as a general mechanism for nonlinear resonance manipulation. Beyond these fundamental aspects, potential applications encompass various fields such as telecommunications, where the data are encoded on a low-frequency modulation applied to a high-frequency carrier, or, more prospectively, microwave signals processing and sensing such as accelerometers featuring enhanced sensitivities via optomechanically-driven signal amplification [21]. Bichromatic signals are in addition pervasive in many other fields including brain-inspired architecture mimicking neural networks [22] where bursting neurons may exhibit two widely different time scales.”

“ ... quantum-limited Josephson parametric amplifiers [32] or, in the optical domain, by phase sensitive amplifiers in the optical domain [33]...”

As requested, and in order to improve and enrich the discussion, we have added new potential outcomes ranging from classical logic gate to quantum computing:

“...We argue that this effect could be used besides the one we are presenting here, as a general mechanism for nonlinear resonance manipulation. Beyond these fundamental aspects, potential applications encompass various fields such as telecommunications, where the data are encoded on a low-frequency modulation applied to a high-frequency carrier, or, more prospectively, microwave signals processing and sensing such as accelerometers featuring enhanced sensitivities via optomechanically-driven signal amplification [21] or for torque magnetometry [Nature Nanotechnology volume 12, pages127–131(2017)]. Bichromatic signals are in addition pervasive in many other fields including brain-inspired architecture mimicking neural networks [22] where bursting neurons may exhibit two widely different time scales. Finally, enhancement of weak signal by vibrational resonance might be valuable for binary logic gate based on phase transition for reprogrammable logic operation [Nature Communications volume 7, Article number: 11137 (2016); Nature Communications volume 2, Article number: 198 (2011)] as well as for memory operation [Nature Nanotechnology volume 6, pages726–732(2011)], physical simulators [Science Advances 24 Jun 2016: Vol. 2, no. 6, e1600236; Scientific Reports volume 6, Article number: 21686 (2016)] and, more prospective application as quantum computing [npj Quantum Information volume 3, Article number: 18 (2017)]”

From the point of view of the key novelties of the manuscript, in the discussion section, a quite extensive discussion is carried out on the theoretical part of the article (see the following sentences) :

“The previous analyses clearly indicate the primordial role of the high frequency amplitude modulation and of the proper timescale separation in such vibrational resonance phenomenon. The former allows to control the nonlinear resonance in order to amplify weak signals. The latter, while being compulsory for technical reasons in the theoretical analysis, could potentially be relaxed in experiments. The exact value of the external drive frequency Ω is not critical at all, as long as it satisfies the timescale separation condition with $\omega_m \ll \omega_f, \Omega$ and if it remains non-resonant. Signal amplification results from the tuning into resonance of the non-linear response of the system. Amplification could occur also in the case of a non-multivalued response, as long as the slope of the tuned response is large enough to ensure appropriate amplification. In principle, for any given signal amplitude, it is always possible to adjust both the forcing strength F and the high frequency modulation strength δ for vibrational resonance to occur.”

On the experimental side, the following sentence has been added :

“We here show an enhancement by a factor up to 20 of a weak modulated signal thanks to vibrational resonance. Moreover, the occurrence of vibrational resonance in a forced system requires the external driving frequency not only to be higher than the signal frequency but also to be lower than the forcing frequency. Most importantly, we show in that case that the high-frequency driving amplitude renormalises the forced nonlinear resonator response through the manipulation of the nonlinear resonance.”

Reviewers' Comments:

Reviewer #2:

Remarks to the Author:

The authors addressed all my previous concerns.

Reviewer #3:

None